# Corrosion Properties of Aluminum Alloy Reinforced with Wood Particles

**Peter Omoniyi [1,2,\*], Olatunji Abolusoro [2], Olalekan Olorunpomi [1], Tajudeen Ajiboye [1], Oluwasanmi Adewuyi [1], Olawale Aransiola [1] and Esther Akinlabi [2,3]**

[1] Department of Mechanical Engineering, University of Ilorin, Ilorin 240003, Nigeria; olorunpomiolalekan@gmail.com (O.O.); ajitek@unilorin.edu.ng (T.A.); adewuyi.oa@unilorin.edu.ng (O.A.); olawalearansiola18@gmail.com (O.A.)

[2] Department of Mechanical Engineering Science, University of Johannesburg, Johannesburg 2006, South Africa; abolusoroolatunji@yahoo.com (O.A.); etakinlabi@gmail.com (E.A.)

[3] Directorate, Pan African University for Life and Earth Sciences Institute (PAULESI), Ibadan 73544, Nigeria

\* Correspondence: omoniyi.po@unilorin.edu.ng or 219126794@student.uj.ac.za; Tel.: +234-813-591-0454

**Abstract:** The need for improved metallic materials in terms of physicomechanical, microstructure, and corrosion properties has necessitated the need to form metal matrix composites. This article adopted the stir casting procedure and used wood particles (WP) to reinforce aluminum alloy at different volume fractions. The corrosion properties of the aluminum matrix in 3.5% wt NaCl were characterized using scanning electron microscopy (SEM), the potentiodynamic polarization curve, and open circuit potential. The corrosion results of the reinforced aluminum alloys was compared with the unreinforced alloy. The unreinforced sample showed improved resistance to corrosion compared to the reinforced aluminum alloy. All samples exhibited visible Pits on SEM observation.

**Keywords:** aluminum; corrosion; NaCl; potentio-dynamic; recycling; wood particles

## 1. Introduction

Aluminum is a lightweight material that has formed an integral part of today's 4.0 industry [1]. Its uses in various industries such as beverages, aerospace, and building have made it indispensable [2–6]. However, improvements are still being made to aluminum and its alloys to better its uses, especially in structural applications. Aluminum and its alloys have been proved to be 100% recyclable. The composites formed from aluminum have been used in manufacturing roofing sheets, electronic packaging applications, and other household items such as aluminum foil due to their good thermal properties and light weight [7–10]. In recent years, aluminum matrix composites (AMCs) have been widely developed, especially with the use of easily sourced agricultural residue as reinforcement [11]. Furthermore, AMCs have been used in microelectronics, bicycles, and golf clubs due to their high resistance to corrosion and light weight [6,11,12]. The selection of fabrication routes has also been one of the criteria for developing aluminum composites. Powder metallurgy, thermal spraying, and hot pressing have been used in producing composite materials [1]. However, the stir-casting procedure has been more accessible and cost-effective in producing aluminum composites [2,13].

Several authors have studied and developed AMCs, and the mechanical properties of these AMCs have been tremendously improved over pure aluminum. Improved physicomechanical properties such as density, which was reduced, and ultimate tensile strength have been increased by reinforcing aluminum alloys with wood particles [2]. Similarly, the use of palm kernel shell ash (PKSA) and silicon carbide (SiC) to reinforce aluminum alloy by Ikubanni et al. [14] has produced improvements in density, hardness, and ultimate tensile strength. The PKSA-reinforced composite exhibited ultimate tensile

strength in the range of 120 MPa compared to that of the WP reinforced composite, which exhibited ultimate tensile strength around 100 MPa. The wear properties of the PKSA- and SiC-reinforced composites were further analyzed by Ikubanni et al. [15]. The reinforced composites show an improved wear resistance compared to the unreinforced samples. Rice husk ash (RHA) has also been a popular reinforcing material used by researchers in improving the tribological and mechanical properties of aluminum alloys [16]. The tensile strength was found at a maximum of 193.79 MPa at 8% addition of RHA, and an improved coefficient of friction was also observed at 8% RHA addition [17]. Despite the improved mechanical properties, adding various agricultural residues created, not all of the reinforcements proved to improve the corrosion properties of aluminum composites.

Intermetallic particles (IMPs) such as iron and manganese have been reported to significantly affect aluminum's alloys' mechanical and corrosion properties due to the higher electrode potential above the aluminum matrix [18]. They have been ascertained to be the source of pitting corrosion in an alkaline environment. Alaneme and Olubambi [19] reported an increase in corrosion rate as the percentage of rice husk increases in the aluminum matrix. A similar study by Ononiwu et al. [20] also showed that an increase in fly ash content reduces the ability of the aluminum alloy to resist corrosion in an alkaline solution. In another study by Ononiwu et al. [21], the addition of fly ash and eggshell, up to 5%, to reinforce aluminum also improves corrosion resistance. Additionally, in the report by Gouda et al. [22], the addition of eggshells to magnesium alloy also improved resistance to corrosion by up to 5%.

The use of reinforcement such as eggshell and fly ash, up to 5% in aluminum alloy, has improved the physicomechanical and corrosion resistance of aluminum. However, in some reports, reinforcements impair the protective oxide film ($Al_2O_3$) formed by aluminum on its surface, resulting in corrosion. In addition, manufacturing processes such as stir casting, machining, and powder metallurgy hamper the aluminum corrosion rate [23]. However, the use of wood particles as aluminum reinforcement has been proved to improve the physicomechanical properties of aluminum alloy [2]. Nevertheless, its corrosion-resistance ability has not been established or widely reported. Additionally, the presence of calcium oxide and silicon oxide in wood particles is believed to improve the corrosion properties of aluminum composite as these compounds have played a significant role in improving the corrosion resistance of eggshell and silicon-carbide-reinforced aluminum composite [21].

Furthermore, works of literature have presented contradictory results on the corrosion properties of aluminum composites. The contradictory results might be due to the different matrix and reinforcement coupled with the composite's production technique. Therefore, this article examines the corrosion properties of aluminum alloy reinforced with wood particles; it also explores the ability to recycle waste materials, thereby turning waste to wealth.

## 2. Materials and Methods

The materials and the method used for the production of samples in this experiment are those of Omoniyi et al. [2], where aluminum waste cans were recycled with wood residue made of hardwood (Mahogany) as reinforcement. The theoretical density of the aluminum alloy is 2.7 g/cm³, and the particle sizes of the wood residue range from 75–125 μm. The chemical composition of the wood particles (WP) and scrap aluminum cans are presented in Tables 1 and 2. Figure 1a,b shows the materials used for the composite.

**Table 1.** Compositional analysis of wood particles.

| Content | $SO_3$ | Volatile | $SiO_2$ | $Al_2O_3$ | CaO | MgO | $Fe_2O_3$ |
|---------|--------|----------|---------|-----------|-----|-----|-----------|
| wt (%) | 0.71 | 21.74 | 60.42 | 3.42 | 8.22 | 3.33 | 2.16 |

**Table 2.** Chemical composition of aluminum alloy. Adapted from [2].

| Element | Si | Fe | Cu | Mn | Mg | Ti | Cr | K | Zn | Al | Others |
|---------|------|------|------|------|------|------|------|------|------|-------|--------|
| wt (%) | 0.59 | 0.43 | 0.07 | 0.39 | 2.14 | 0.01 | 0.01 | 0.01 | 0.19 | 96.04 | 0.11 |

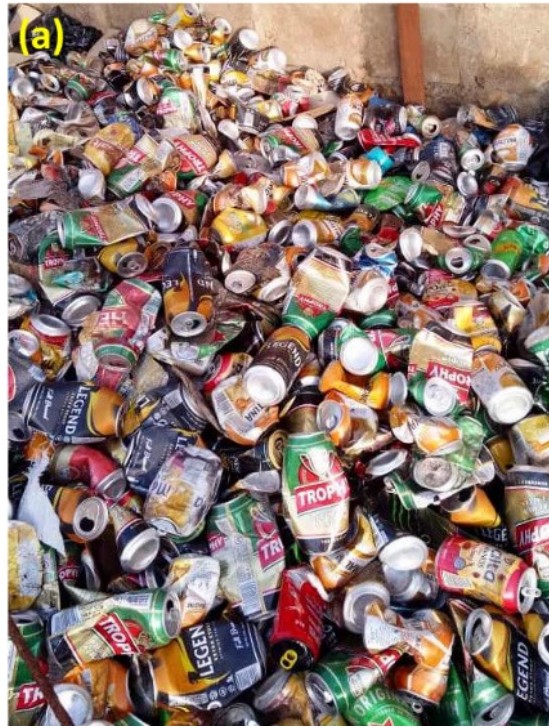
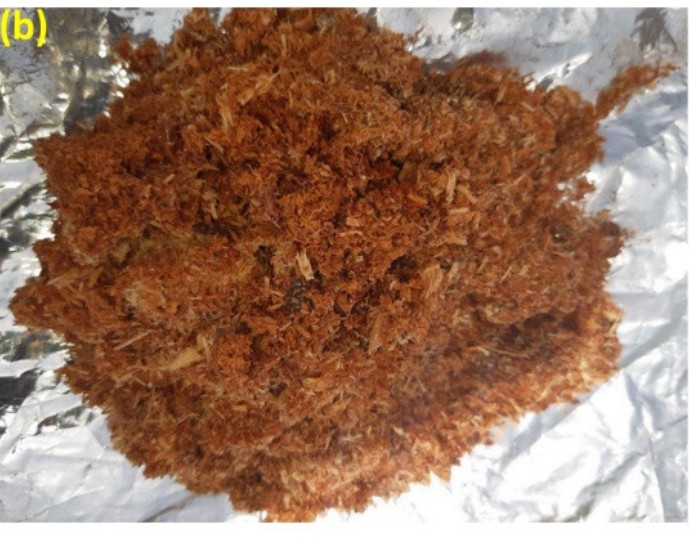

**Figure 1.** (**a**) Aluminum scraps; (**b**) wood particles.

### 2.1. Production of Composite and Characterization

The stir casting technique was adopted in the production of the composite material. The aluminum alloy was placed in a crucible and charged into a furnace preheated to 500 °C. The temperature in the furnace was further increased to 800 °C for 45 minutes to let the aluminum alloy melt completely. The crucible was brought out of the furnace to remove impurities and slags, and then the wood particles (WP) ranging from 75–125 μm were charged into the crucible. The composite was returned to the furnace and stirred for 20 minutes at 450 rpm. The composite was then poured into a prepared mold and allowed to cool at room temperature. Figure 2 shows the process route to produce the aluminum composite. Additionally, the experimental matrix is presented in Table 3.

**Table 3.** Alloy matrix composition.

| SN | Al Alloy Composition (% wt) | WP (% wt) |
|----|------------------------------|-----------|
| 1 | 100 | 0 |
| 2 | 85 | 15 |
| 3 | 80 | 20 |

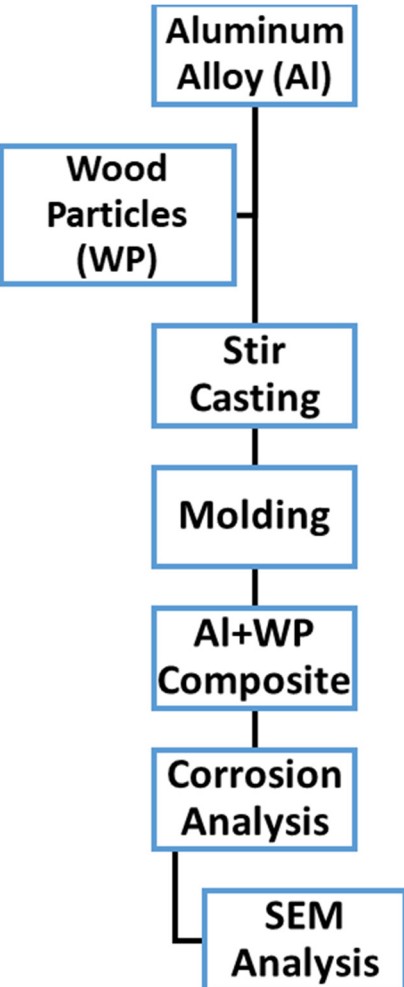

**Figure 2.** Composite production route.

### 2.2. Corrosion Specimen Preparation

Three sets of samples measuring 10 mm in diameter and 3 mm thick were cut out of the cast sample for corrosion analysis. Each sample underwent grinding using SiC papers (#320–#4000) and was polished until a mirror surface was achieved. Copper wire was attached to each sample, and the surfaces not polished were covered with epoxy resin to avoid the exposure of the unpolished parts to the corrosion fluid. Ag/AgCl platinum rods were used as counter electrodes, and the sample was set as the working electrode. The DY2300 model potentiostat was used in measuring the potentials of the aluminum alloy. The polished surface was exposed to 3.5% wt NaCl at room temperature for two hours to obtain a stable open circuit potential (OCP), −0.25 V to +0.25 V was used as the dynamic potential, and 1.0 mV/s was used as the potentiodynamic scan rate. Other parameters measured were the potential corrosion $E_{corr}$ (V) and the corrosion current $I_{corr}$ (A).

## 3. Results and Discussion

### 3.1. Electrochemical Analysis of Composite

Figure 3 shows the open circuit potential (OCP) plot, which is the potential in the composite material when no current is applied in the system [24,25]. From the plot, each sample shows variations in potential with respect to exposure time, which could result from pores and inhomogeneity, as explained by Yadav et al. [26]. It could further be attributed to simultaneous corrosion product formation and breakdown, described by Alaneme and Olubambi [19]. The unreinforced aluminum alloy had the most stable OCP value, indicating more thermodynamic stability than the other samples. In contrast, 20%

wood particle addition had the least stable OCP values, indicating the possibility of it being the most thermodynamically unstable and being able to corrode faster than other samples, which could be a result of the inhomogeneity of the composite. The downward trend of the OCP of the 20% wood particle addition could also be attributed to the loss of passivity due to the thinning of the oxide layer of aluminum oxide ($Al_2O_3$), whereas the upward trend of the addition of 15% wood particles until about 600 s could be due to the repassivation of the oxide layer protecting the surface of the composite. Generally, the fluctuations of the OCP values through the period of 1800 s could be further due to incessant passivation and repassivation as a result of the exposure of the composites to a corrosive environment [27].

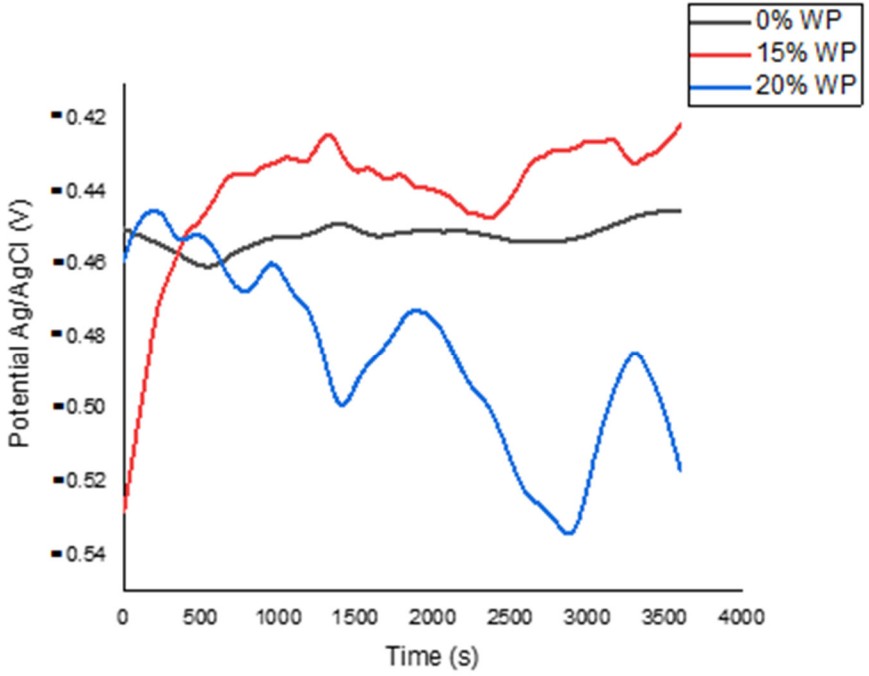

**Figure 3.** Open circuit potential (OCP) plot of the composite.

Figure 4 shows the Tafel plot, where all samples show a similar polarization curve as the cathodic and anodic curve patterns follow the same trend. In Table 4, the unreinforced sample shows improved corrosion resistance as the current density ($I_{corr}$) values are lower than the reinforced aluminum composites. According to Faraday's law, corrosion current density has a linear relationship with the corrosion rate [26], which results in the unreinforced composites having a lower corrosion rate, that is, the higher current density value tends to be more susceptible to pitting corrosion. The increase in current density as the percentage of wood particles increases in the composite could be due to impurities and inclusions such as silicon, and carbon, which results from the addition of wood particles. Researchers have reported that these inclusions make aluminum-based composites more susceptible to pitting corrosion attacks [28]. Generally, all samples had a corrosion rate within the very stable condition of $10^{-3}$ mm/yr, as stipulated by Yang et al. [29].

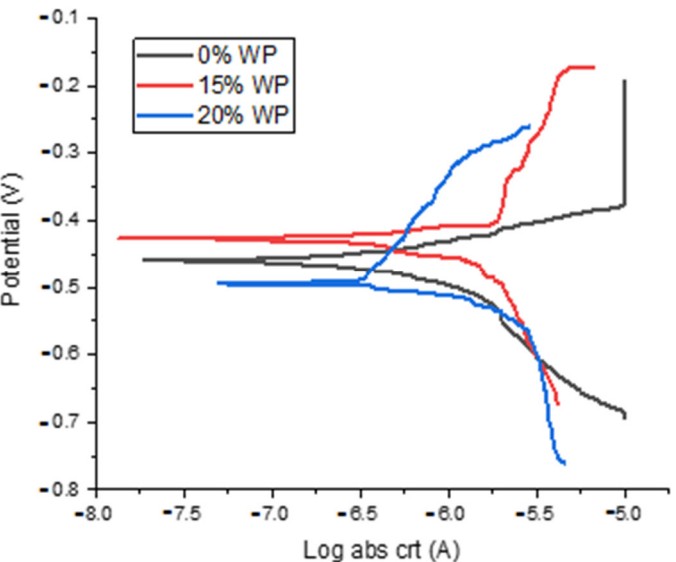

**Figure 4.** Tafel curve of the composites.

**Table 4.** Potentiodynamic curve (Tafel plots) parameters.

| Sample | $I_{corr}$ ($\mu A/cm^2$) | $E_{corr}$ (V) | Rct (ohms) |
|---------|---------|---------|---------|
| 0% WP | 0.712 | −0.459 | $3.610 \times 10^4$ |
| 15% WP | 1.220 | −0.426 | $2.105 \times 10^4$ |
| 20% WP | 1.767 | −0.495 | $1.454 \times 10^4$ |

*3.2. Scanning Electron Microscope (SEM) Analysis of Composite*

Figures 5–7 show the composites' scanning electron micrographs (SEM). The observations show clusters of Pitt corrosion attacks on the surface of each sample after exposure to the 3.5% wt NaCl solution. The formation of pits on the surface of the samples further shows that the passive film ($Al_2O_3$) formed against corrosion attack was not enough to sustain the effect of the halide ions ($Cl^-$) [21,23,30]. It could also be due to inclusions and intermetallic formed within the composites [18,26]. From the EDS charts, the presence of oxygen (O), chlorine, sodium, and carbon in the composites suggested the formation of aluminum carbide, which Alaneme et al. [31] explained is responsible for the adverse effect of corrosion in aluminum composites. However, the presence of silicon carbide from the wood particles was not sufficient to mitigate the effect of the $Al_4C_3$ formation in the composite. The reaction of aluminum and carbon is presented in Equation (1). Furthermore, the reaction of aluminum with the chloride ions ($Cl^-$) generated from the dissolution of sodium chloride (NaCl) in water to form aluminum chloride ($AlCl_3$), which results in pitting corrosion on the metal surface, is presented in Equation (2) [32].

$$4Al + 3C = Al_4C_3 \tag{1}$$

$$Al + 3Cl = AlCl_3 \tag{2}$$

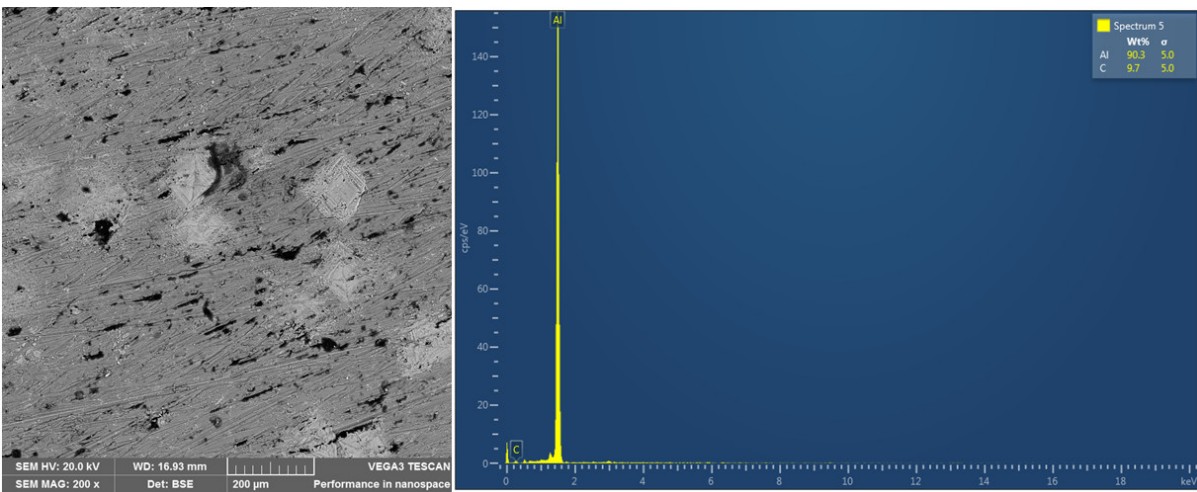

**Figure 5.** Scanning electron micrograph and EDS chart of 0% wood-particle-reinforced composite.

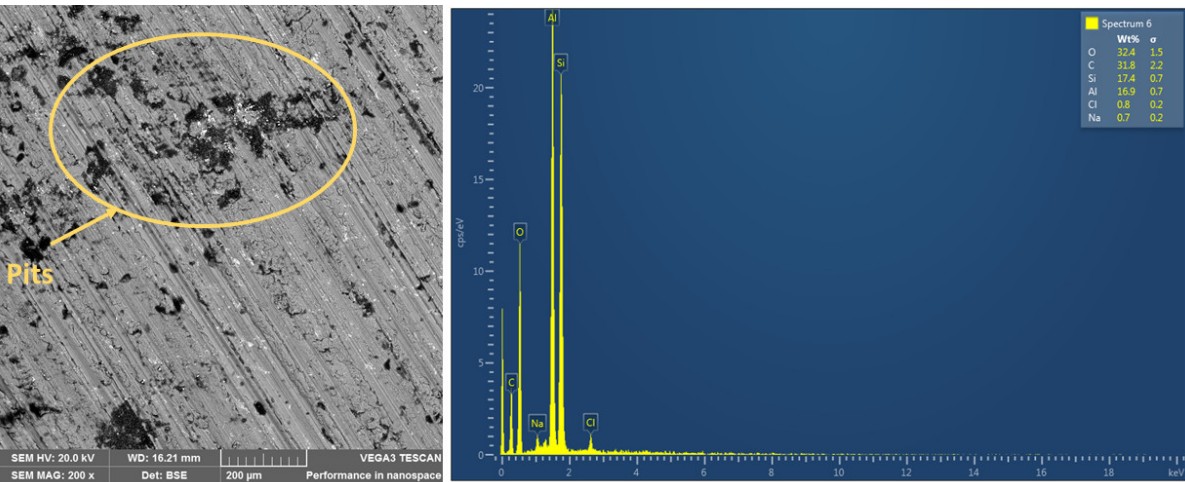

**Figure 6.** Scanning electron micrograph and EDS chart of 15% wood-particle-reinforced composite.

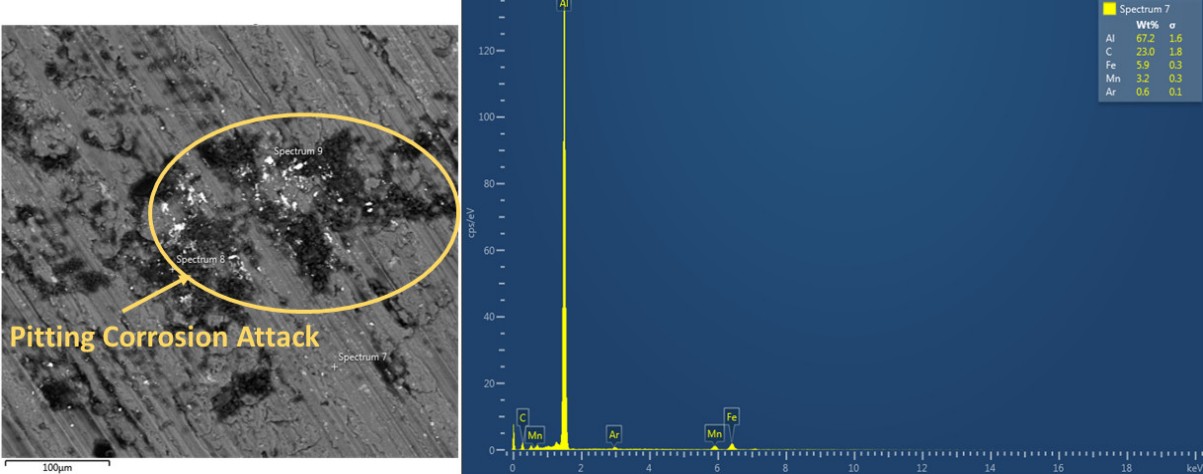

**Figure 7.** Scanning electron micrograph and EDS chart of 20% wood-particle-reinforced composite.

### 3.3. X-Ray Diffraction (XRD) Analysis of Composite

Figure 8 shows the X-ray diffraction (XRD) analysis done on the composites to establish the phase and existence of reinforcement in the samples. The prominent crystal planes observed are Al (111) and Al (200), of which (111) is more prominent, except for that of 15% wood particle addition. The phases confirm the face-centered cubic (FCC) nature of the crystals. The phases were consistent for all samples, which denotes that no new phase was developed during the manufacturing process due to the wood particles getting burnt as volatile matter during the manufacturing process or the fraction volume of the wood particles being small and undetectable by XRD, except for the 20% wood particle addition sample, where traces of aluminum carbide were detected. The peaks verified the existence of aluminum, as also observed by El-Fattah et al. [22] and Ikubanni et al. [33], where eggshell was used as reinforcement for aluminum.

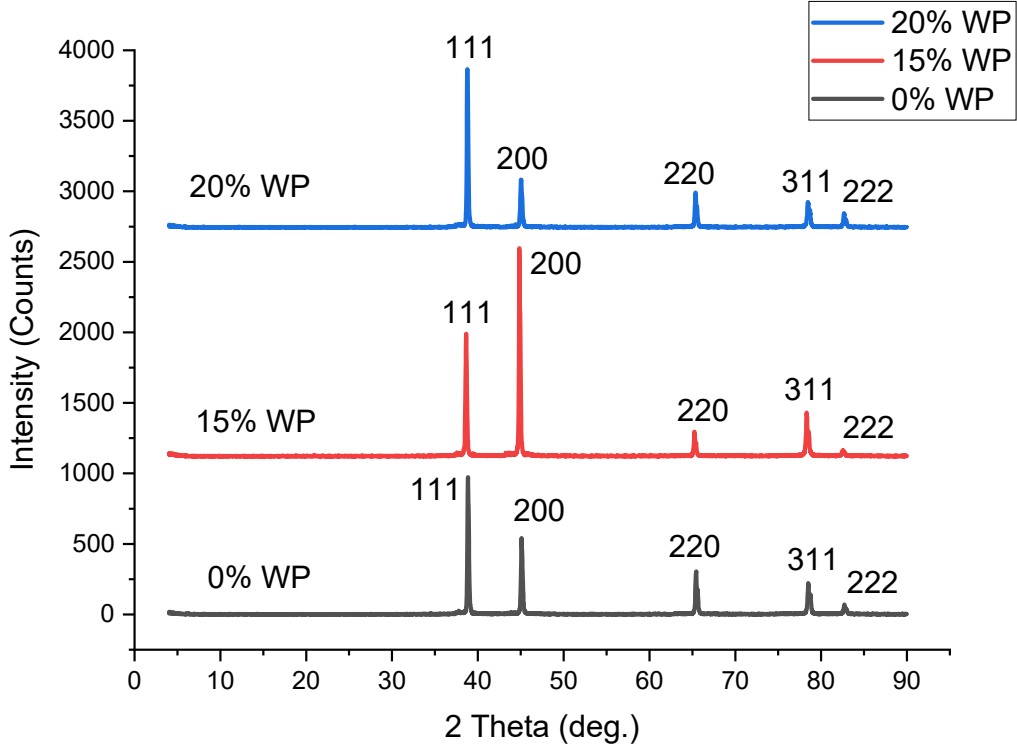

**Figure 8.** XRD pattern of composite material.

### 4. Conclusions

Aluminum-based composites reinforced with wood particles have been produced through stir casting, and the corrosion properties of the composite have been studied. The following conclusions are drawn from the study:

1. The as-received aluminum alloy was less susceptible to corrosion than the produced composites due to little or no inclusions in the microstructure of the material, whereas the aluminum composite exhibited a substantial level of pitting corrosion from the SEM images observation as a result of the formation of aluminum carbide formation within the composite

2. The composites exhibited a corrosion rate within the very stable condition of $10^{-3}$ mm/year.

3. The corrosion rate increased as the amount of wood particle reinforcement increased in the composite. An increase of 42% in current density was observed as wood particles increased to 15%, and a further increment of 31% was observed as the wood particles increased to 20%.

Even though the presence of silicon and oxygen in the elemental composition of wood particles is to mitigate the rate of corrosion of the composite through the formation of silicon carbide (SiC) and alumina ($Al_2O_3$), the presence of aluminum carbide had a more significant effect in terms of increasing the corrosion rate of the composite. The authors recommend reducing the aluminum carbide level in future research.

**Author Contributions:** Conceptualization, P.O. and O.O.; methodology, P.O., O.O. and O.A. (Olatunji Abolusoro); software, E.A. and P.O.; validation, O.A. (Olatunji Abolusoro), E.A., T.A., O.A. (Oluwasanmi Adewuyi), and O.A. (Olawale Aransiola); formal analysis, P.O.; investigation, P.O. and O.O.; resources, E.A., P.O. and O.O.; data curation, P.O. and O.O.; writing—original draft preparation, P.O.; writing—review and editing, E.A., O.A. (Olatunji Abolusoro), O.A. (Oluwasanmi Adewuyi), O.A. (Olawale Aransiola) and T.A.; visualization, P.O. and O.A. (Olatunji Abolusoro); supervision, P.O., O.A. (Olatunji Abolusoro), E.A., O.A. (Oluwasanmi Adewuyi), O.A. (Olawale Aransiola) and T.A.; project administration, P.O., E.A. and T.A.; and funding acquisition, E.A. All authors have read and agreed to the published version of the manuscript.

**Funding:** This research received no external funding.

**Institutional Review Board Statement:** Not applicable.

**Informed Consent Statement:** Not applicable.

**Data Availability Statement:** The data presented in this study are available on request from the corresponding author.

**Acknowledgments:** The authors acknowledge the workers at the foundry workshop, Department of Materials and Metallurgical Engineering, University of Ilorin for their support during the composite manufacturing.

**Conflicts of Interest:** The authors declare no conflict of interest.

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
