# Peer review of "Corrosion Properties of Aluminum Alloy Reinforced with Wood Particles"

_jcs, doi:10.3390/jcs6070189_

Round 1
Reviewer 1 Report
Congratulations. The work is on track, however, there are some concerns about your work that can be addressed by improving the work and your understanding. Please, see the attachment.

Reviewer 2 Report
- In the Introduction section, the authors cited the specific results of previous research and cited them adequately. However, they did not mention their shortcomings in previous research. In the Introduction section, the penultimate paragraph should contain common features of previous research. The shortcomings of previous research should also be pointed out, in general.
- In the Introduction section, the last paragraph should contain the scientific contribution and scientific hypotheses of your research. Complete, further elaborate the scientific contribution and scientific hypotheses of your research. Be explicit. In addition to the goal of the research (which was written), the novelty in the context of the scientific contribution should be pointed out. Scientific contributions should be written based on the shortcomings of previous research in the literature. In this way, the authors will better emphasize novelty and scientific soundness.
- Analyze and discuss possibilities of practical application.
- Figure 1 a , improve the quality of the image.
- In the conclusions, state the scientific contribution, the shortcomings of your methodology and future research.
- Generally, the quality of the writing could be improved.
Reviewer 3 Report
1. Line 30: Citation missing.
2. what kind of wood was used? And was it dispersive when mixed with aluminum?
3. line 102: what is epoxy resin used for?
4. a conductive metal is susceptible to corrosion. Isn't it more logical that addition of wood particles lowers the electrical conductivity and thereby the reduced corrosion rate? Why does addition of wood particle increase the corrosion rate?
5. It is recommended to identify the relevant mineralogy in XRD figure 8. Authors can employ the open source softwares with vast database to identify them.
6. Is it 10-3 mm/yr.? or is there is a typo 1.0-3 mm/yr? or should it be 3-10mm/yr.?
7. In 0% WP, what type of corrosion was observed? Pitting or uniform?
Round 2
Reviewer 1 Report
With the changes introduced in the work, the authors improved the article, reaching the necessary quality for its publication.
Congratulations.
Reviewer 2 Report
The article has been significantly improved
Reviewer 3 Report
The response of the authors are found to be satisfactory and the revised manuscript is recommended for publication.